# Identification of Drug Repurposing Candidates for Coxsackievirus B3 Infection in iPSC-Derived Brain-like Endothelial Cells

**DOI:** 10.3390/ijms26157041

**Published:** 2025-07-22

**Authors:** Jacob F. Wood, John M. Vergis, Ali S. Imami, William G. Ryan, Jon J. Sin, Brandon J. Kim, Isaac T. Schiefer, Robert E. McCullumsmith

**Affiliations:** 1Department of Neurosciences and Psychiatry, University of Toledo College of Medicine, Toledo, OH 43699, USA; jacob.wood@rockets.utoledo.edu (J.F.W.); john.vergis@rockets.utoledo.edu (J.M.V.); ali.imami@rockets.utoledo.edu (A.S.I.); william.ryan2@rockets.utoledo.edu (W.G.R.V); 2Department of Biological Sciences, University of Texas at Dallas, Richardson, TX 75080, USA; jon.sin2@utdallas.edu (J.J.S.); brandon.kim@utdallas.edu (B.J.K.); 3Department of Medicinal and Biological Chemistry, University of Toledo College of Medicine, Toledo, OH 43614, USA; isaac.schiefer@utoledo.edu; 4Neuroscience Institute, ProMedica, Toledo, OH 43606, USA

**Keywords:** differential gene expression, Coxsackievirus B3, blood–brain barrier, transcriptomics, drug repurposing

## Abstract

The enterovirus Coxsackievirus B3 causes a range of serious health problems, including aseptic meningitis, myocarditis, and pancreatitis. Currently, Coxsackievirus B3 has no targeted antiviral treatments or vaccines, leaving supportive care as the primary management option. Understanding how Coxsackievirus B3 interacts with and alters the blood–brain barrier may help identify new therapies to combat this often-devastating infection. We reanalyzed a previously published RNA sequencing dataset for Coxsackievirus B3-infected human-induced pluripotent stem-cell-derived brain endothelial cells (iBECs) to examine how Coxsackievirus B3 altered mRNA expression. By integrating GSEA, EnrichR, and iLINCs-based perturbagen analysis, we present a novel, systems-level approach to uncover potential drug repurposing candidates for CVB3 infection. We found dynamic changes in host transcriptomic response to Coxsackievirus B3 infection at 2- and 5-day infection time points. Downregulated pathways included ribosomal biogenesis and protein synthesis, while upregulated pathways included a defense response to viruses, and interferon production. Using iLINCs transcriptomic analysis, MEK, PDGFR, and VEGF inhibitors were identified as possible novel antiviral therapeutics. Our findings further elucidate Coxsackievirus B3-associated pathways in (iBECs) and highlight potential drug repurposing candidates, including pelitinib and neratinib, which may disrupt Coxsackievirus B3 pathology at the blood–brain barrier (BBB).

## 1. Introduction

Coxsackievirus B3 (CVB3) is a non-enveloped, single-stranded RNA virus that belongs to the Enterovirus B species within the Picornaviridae family [1]. Enteroviruses, including CVB3, are among the leading causes of aseptic meningitis in neonates, accounting for 85–95% of all cases [2]. The highest incidence (approximately 75%) of CVB1-B5 infection occurs in infants under 1 year old and toddlers from 1 to 4 years old, which is attributed in part to poor hygiene practices among infants [3]. In addition to CVB3’s role in neonatal disease, CVB3 also causes viral myocarditis, respiratory and musculoskeletal syndromes, as well as dermatologic manifestations with varying severity of disease in adults [3,4]. Non-polio enteroviruses are responsible for 10–15 million infections in the United States alone, and from the years 1970 to 2005, about 5% of fatal enterovirus cases were identified as CVB3 in origin, while 25% of enteroviral infections in the United States were reported to be CVB1-B5 [5,6].

Despite its pathological impact on the central nervous system (CNS), CVB3 infection has no approved targeted antiviral therapies or available vaccines [7]. Treatment for Coxsackievirus syndromes, such as viral meningitis and hand, foot and mouth disease, remains primarily supportive, consisting of fluid management, pain control, and, in severe cases, inpatient hospitalization [8].

While systemic effects of CVB3 are well documented, CVB3 interaction with the blood–brain barrier (BBB) remains poorly understood. The pathogenesis of CVB3 in the CNS raises important questions, especially given the virus’s known ability to infect a wide range of cell types and tissues, with known target cell types in the CNS (Table 1) [9,10,11,12,13]. Recent transcriptomic analyses have shed light on how CVB3 alters transcriptomic signatures in brain endothelial cells (BECs), which remain an important cell type to evaluate, given the devastating effects that CVB3 has on the CNS [14]. As the primary cellular interface between the bloodstream and CNS parenchyma, BECs play a pivotal role in the regulation of viral entry, making them a critical in vitro model to investigate CVB3 CNS entry and infection [15,16,17].

In this study, we reanalyzed a previously published RNAseq dataset (GSE269413) examining CVB3 infection in human-induced pluripotent stem-cell-derived brain endothelial cells (iBECs) at two- and five-day post-infection (dpi) time points [14]. While the original study examined pathways and transcriptomic signatures at both time points, we extended the findings using advanced pathway and drug repurposing analyses. We employed cutting-edge transcriptomic analysis using GSEA, EnrichR, and iLINCs, with a focus on identifying drug leads that reverse the CVB3-associated transcriptomic signatures in iBECs (Figure 1) [18]. Our findings provide improved insight into potential antiviral strategies and therapies for CVB3-associated CNS disease.

## 2. Results

We visualized significantly differentially expressed genes (*p*adj < 0.05) of 2 and 5 dpi using volcano plots and heatmaps (Figure 2). At 2 dpi, there were 444 downregulated and 644 upregulated genes, while 5 dpi showed 3956 upregulated and 4475 downregulated genes. When applying a log2FC filter threshold of >2.0, 13 upregulated and 25 downregulated genes were found at 2 dpi, and 403 upregulated and 164 downregulated genes were found at 5 dpi (Figure 2). The most significantly upregulated gene at 5 dpi was OAS2 (Log2FC: 10.28), which was also upregulated at 2 dpi (Log2FC: 1.79) (Figure 2). *TREML2* was the most downregulated gene in the 2 dpi (Log2FC: −4.66) (Figure 2). *CGA* was the most downregulated gene shared at 2 (Log2FC: −4.43) and 5 dpi (Log2FC: −1.80) (Figure 2).

Pathway analysis of 2 dpi using GSEA identified 629 significantly altered pathways, with 549 upregulated and 80 downregulated. The top 10 upregulated pathways included “negative regulation of viral genome replication,” “defense response to virus,” and “interferon-beta production” (Table 2). The top 10 downregulated pathways included “hormone metabolic process,” “cell–cell fusion,” and “syncytium formation” (Appendix A). Pathway analysis of 5 dpi using GSEA identified 1172 significantly altered pathways, with 930 upregulated and 242 downregulated. The top 10 upregulated pathways included “response to virus,” “response to external biotic stimuli,” and “innate immune response” (Table 2). The top 10 downregulated pathways included “cytoplasmic translation,” “ribosomal subunit,” and other ribosomal-related pathways (Appendix A).

Next, we performed leading-edge gene analysis using GSEA of 2 and 5 dpi, which showed *IL6* and *TNF* as the top upregulated leading-edge genes (Appendix A). *IL6* and *TNF* contributed to 313 (Appendix A) and 530 pathways (Appendix A), respectively. No shared genes were found in the top 10 downregulated leading-edge genes between 2 and 5 dpi. *AGTR1* was found to be the top-bottom leading edge gene at 2 dpi (N = 15) (Appendix A). Furthermore, *SCN5A* was found to be the top-bottom leading edge gene at 5 dpi (N = 44) (Appendix A).

Pathway analysis of 2 dpi using EnrichR identified 724 significantly altered pathways, with 530 pathways upregulated and 194 pathways downregulated. The top upregulated EnrichR pathways included interferon signaling, i.e., “Cellular response to type I interferon” as well as “regulation of ribonuclease activity” (Appendix A). The top downregulated EnrichR pathways included “retinal cone development” and “negative regulation of morphogenesis of an epithelium,” though only “muscle myosin complex” was significant in the top 10 list (Appendix A). Pathway analysis of 5 dpi using EnrichR identified 1289 significantly altered pathways, with 1000 pathways upregulated and 289 pathways downregulated. The top pathways included regulation of viral processes, i.e., “Cellular response to type I interferon” and “MHC class antigen processing,” which were upregulated (Appendix A). The top downregulated pathways included “cellular response to potassium” as well as other calcium channel-related pathways, i.e., “Regulation of smooth muscle contraction” (Appendix A).

Perturbagen analysis of the 2 dpi using iLINCs found discordant chemical compounds associated with MEK pathway inhibition and EGFR inhibitors, including selumetinib (discordance score: −0.46) and tyrphostin AG-1478 (−0.492), respectively (Table 3). The most discordant drug in the 2 dpi perturbagen list was N6022, which had no annotated mechanism of action in iLINCs, and was found to be a small-molecule GSNOR inhibitor (discordance score: −0.500) (Table 3). Hydroquinidine was the most discordant 5 dpi perturbagen (discordance score: −0.464) (Table 4). The discordant perturbagen list at 5 dpi included kinase and receptor-focused compounds such as MTOR, FGFR, and IGF-1 inhibitors, including AZD-8055 (discordance score: −0.457) (Table 4). The small molecule MLS000718723 (concordance score: 0.70) was found to be the top concordant perturbagen at 2 dpi, while SB-218078 (concordance score: 0.54) was found to be the top concordant perturbagen at 5 dpi (Appendix A).

“VEGFR inhibitors” were highly represented in iLINCs perturbagen MOAs as both concordant and discordant signatures, with 88 concordant hits for the 2 dpi, and a combined 20 discordant hits at 2 dpi (Appendix A). “PDGFR inhibitors” were also highly represented as iLINCs perturbagen discordant MOAs, with 16 hits for 2 dpi, and 24 hits for 5 dpi (Appendix A). The top concordant perturbagen MOA at 5 dpi was CDK inhibitors, with 20 concordant hits (Appendix A).

## 3. Discussion

Our analysis shows a time-dependent shift in transcriptomic profiles following CVB3 infection of iBECs at 2 and 5 dpi, highlighting immune activation, pathway dysregulation, and potential therapeutics (Figure 3). Utilizing the “3PodR” transcriptomic toolkit, we confirmed many of the results in the original study, such as the involvement of responses to virus pathways and interferon signaling, but we extended the original findings to introduce cutting-edge pathway and perturbagen analysis [19]. Our use of GSEA and EnrichR within 3PodR (https://github.com/willgryan/3PodR_bookdown.git; accessed on 8 August 2024) also allows for identifying mechanistic signaling pathways (e.g., KEGG, Reactome) based on gene set enrichment. While Gene Ontology (GO) enrichment often shares biological functions or processes, pathway-based analyses such as GSEA can offer more direct insights into signaling cascades and molecular mechanisms relevant to disease states [20,21].

At 2 dpi, we observed an early inflammatory response, with upregulation of interferon pathways, including *IFIT1* and *IL6* genes (Figure 2). By 5 dpi, we found an intensified host cell immune response, including the upregulation of *IFNB1*, *TNF*, and *OAS2*, all important viral response markers [22,23,24]. Notably, *OAS2* plays a critical role in RNA degradation via the RNase L pathway and indicates an intense viral state [25]. This aligns with clinical findings that children who harbor *OAS2* polymorphisms have increased susceptibility to enterovirus infection [26]. Furthermore, the downregulation of ribosomal biogenesis genes at 5 dpi suggests that the virus may be dampening host protein synthesis to subvert immune signaling, which is consistent with the known strategy of enteroviruses to hijack and shut down host translation machinery via viral-induced particles and enzymes (Appendix A) [27]. The temporal shift in genetic expression highlights a biphasic infection state, where immune signaling becomes activated, and cellular dysregulation may occur through ribosomal pathways. This immune-related biphasic response is highlighted by the differential upregulation of mRNA expression among shared GSEA pathways at 5 dpi versus 2 dpi (Figure 3A,B). It should be noted that the lack of professional immune cells and systemic feedback loops in an iBEC model system could be responsible for the changes in amplitude of gene expression. Without the support of microglia, perivascular macrophages, circulating leukocytes, or complement factors, the time dependent shifts in transcriptomic signatures may vary based on higher order support systems in vivo.

Leading-edge genes are essentially “prominent” genes within a set of pathways that contribute a high-ranking score to that pathway. A gene that is more common in many of the leading-edge gene lists may be more significant than those listed less frequently. This can help us to identify specific genes of interest that contribute the most to pathway analysis, helping to narrow a targeted gene search. Leading-edge gene analysis further supports the role of inflammatory processes in CVB3 infection. Notably, *TREML2* and *CGA* emerged as the top downregulated leading-edge genes at 2 dpi. CGA is a well-studied anti-inflammatory/antiviral compound, whereas *TREML2* is implicated in microglial modulation and neuroinflammation. Genes such as *CCL4L2* appeared as the top upregulated leading-edge genes at 5 dpi. *CCL4* is a crucial chemokine for macrophage signaling and inflammatory processes [28,29,30,31]. *IL6* and *TNFα* also emerged as top leading-edge genes at 2 and 5 dpi, contributing to over 313 pathways (Appendix A). These findings highlight a viral-mediated cytokine and inflammation narrative of CVB3 infection, enhancing inflammation in iBECs and likely contributing to BBB disruption in vivo.

Using iLINCs’ perturbagen analysis, many experimental and FDA-approved drugs and small molecules producing discordant and concordant scores for CVB-infected iBECs were identified at 2 and 5 dpi. MAPK pathway inhibition is a hallmark of our top ten discordant lists for 2 dpi. Notably, the original study also found that the MAPK signaling pathway was activated during CVB3 infection through Gene Ontology analysis (GO) [14]. Additionally, the authors showed that treatment with the MEK1/2 inhibitor U0126 increased viral protein accumulation [14]. Drug hits such as Selumetinib, Trametinib, PD-0325901, CL-1040, and AZ628 are all present in the discordant list, and all have documented targets of MAP2K and/or MEK1/2. The presence of MAPK inhibitors reversing a CVB3-induced transcriptomic signature is consistent with previous studies, which found that enteroviruses activate MAPK/ERK signaling during early entry [32]. This warrants the investigation of the identified MAPK inhibitors as therapeutic agents for CVB3 infection, given the proposed role of MAPK in CVB entry or exit.

Among the most interesting findings in our iLINCs analysis, Vascular Endothelial Growth Factor Receptor (VEGFR) inhibitors were highly enriched as discordant perturbagens at 2 dpi (Appendix A), suggesting a mechanism to reverse CVB3-induced changes in endothelial cells. However, VEGFR inhibitors were also present as one of the top concordant perturbagen MOAs at 2 dpi (Appendix A). This temporal difference reflects changes in the cellular response during the progression of CVB3 infection. At 2 dpi, VEGFR inhibitors may interact with a more dynamic VEGF signaling environment, while at 5 dpi, the transcriptional profile of infected iBECs appears more consistently reversed by EGFR and PDGFR inhibition, suggesting a possible therapeutic benefit (Appendix A). Previous studies found that VEGF inhibitors such as pazopanib decrease enterovirus infection in a dose-dependent manner and produce global anti-enterovirus effects in cell culture [33]. Our findings extend these observations by highlighting that modulation of growth factor receptor signaling, particularly EGFR, PDGFR, and VEGFR, may offer therapeutic benefit in reversing the CVB3-infected iBEC transcriptome during the later stages of infection.

The therapeutic theme of Growth Factor Receptor inhibitors can also be visualized by a similarity matrix of the top 22 discordant drugs at 2 dpi when cross-analyzing perturbagen similarity scores. The 2 dpi time point was chosen given the therapeutic relevance of early identification and intervention of CVB3 infection. Neratinib and pelitinib both emerge within a molecular similarity cluster, sharing similarity scores of 0.75 (Figure 4A), reflecting the shared core structure, which only varies at the para position of the proximal ring to the quinoline core (Figure 5A). Pelitinib and neratinib are structurally related EGFR inhibitors and may interfere with CVB3-mediated disruption of growth signaling pathways in endothelial cells. Visualization of chemical similarity with a UMAP plot (Figure 4B,C) shows clustered molecular similarity for pelitinib and neratinib, as expected, as well as related compounds, namely erlotinib and tyrphostin_AG-1478-, based on the Tanimoto coefficient of these drugs. Tanimoto coefficients closer to “1.0” indicate higher chemical structure similarity. The two other pharmacophoric chemotypes that emerge from the cluster analysis are the RTK inhibitors Regorafenib and Rebastinib (Figure 5B), as well as several MEK1/2 inhibitors which contain a similar biaryl amine-linked halogenated substructure, most of which also contain a hydroxamic linker, except for Trametinib (Figure 5C).

We also examined the binding properties of the drugs that were identified, including neratinib, pelitinib, erlotinib, and tyrphostin AG-1478 (Figure 5). Though not FDA-approved for therapeutic use, tyrphostin AG-1478 has been implicated in reducing the phosphorylation of MAPK and c-Src/EGFR caused by enterovirus-71, highlighting the inclusion in our docking simulation [34]. Using SwissTargetPrediction, we identified 18 shared gene targets for which these four compounds had listed binding probabilities [35] (Figure 5D,F). We also searched the top iLINCs drug hits for EGFR and MEK1/2 inhibitors (Figure 5E) Pelitinib and Neratinib shared the most similar gene targets, registering 60 similar hits (Figure 5D). Gene targets such as SRC, LCK, and EGFR were found to have the top binding probabilities across all four drugs, with binding probability scores > 0.9823 for each drug (Figure 5F). Interestingly, previous studies have shown the LCK containing pathways are important for CVB viral replication in cardiac and immune cells [36]. As a top shared target for the four discordant drugs this study identified, this could be a promising area of investigation for future studies.

As another potential therapeutic for CVB3 infection, we identified PDGFR inhibitors in our iLINCs analysis. Though their mechanism of interaction remains unclear, PDGFR inhibitors were discordant to transcriptomic signatures at 2 and 5 dpi (Appendix A). The PDGFR pathway is implicated in CVB myocarditis, with upregulation of PDGF pathways in areas of inflammatory fibrosis in cardiac tissue [37]. Enterovirus 71 induces Cox-2 expression in rat brain astrocytes through the expression of the PDGFR pathway [34]. Cox-2 is implicated in neurodegeneration and inflammation, likely as a neurotoxic agent [38]. Taken together, these findings suggest that PDGFR inhibitors may interfere with CVB3 pathogenesis. The mechanism of PDGFR inhibitors in CVB3 infection remains speculative but is a promising lead for further investigation.

iPSC-derived blood–brain barrier endothelial cells (BECs) share many key features with native BECs, such as tight junctions, nutrient transporters, and efflux activity [39]. Native BECs, which develop from the mesoderm and are shaped by in vivo signals, are more mature and responsive to various stimuli [40]. Though they are of mesoderm origin, iPSC-derived BECs only partially replicate the functional maturity or barrier integrity of their native counterparts. iPSC-BECs also exhibit differences in transcriptional profile, affecting their transport functions and signaling activity. However, despite these differences, iPSC-derived BECs still provide a valuable model for studying the blood–brain barrier, particularly in areas like drug development and disease modeling [41,42,43].

This study has several limitations. First, the iBECs used in the study are derived from a female iPSC line, which limits assessment of sex effects. More sex-specific studies are needed to explore the differences and similarities during CVB3 infection. Alterations in mRNA transcription may reflect changes in gene expression; however, these do not necessarily correlate with functional protein activity within the cell due to factors such as host-mediated mRNA degradation and post-transcriptional modifications, as shown in previous studies [44]. Lastly, because these cells are derived from iPSCs and are maintained entirely in vitro, this model does not fully replicate in vivo conditions.

## 4. Materials and Methods

Data analysis overview: We re-analyzed a published GEO dataset (GSE269413) using cutting-edge bioinformatic tools [18,19,20,45]. The original study examined CVB3 infection at 2 and 5 dpi in induced pluripotent stem-cell-derived brain endothelial cells (iBECs). Raw read transcriptomic data were downloaded from supplementary files available on the Gene Expression Omnibus (GEO) [14]. We deployed differential gene expression analysis, pathway analysis using GSEA and EnrichR, and perturbagen analysis using iLINCs. These features are incorporated into an R code composite package, termed “3PodR” (https://github.com/willgryan/3PodR_bookdown.git; accessed on 8 August 2024) [19].

Differential Gene Expression Analysis: Differential gene expression was performed via a limma-voom composite R script, making the comparison of CVB3-infected iBECs vs. mock-infected iBECs at both 2 and 5 dpi [46,47]. An adjusted *p*-value threshold (*p*adj) of <0.05 was established using a Benjamini–Hochberg adjustment. The Python package matplotlib (v3.9.0) was utilized to create a differentially expressed genes (DEG) volcano plot using adjusted *p*-values and log2 fold change in genetic expression derived from the limma-voom output (v3.5.0) [48]. The adjusted *p*-value threshold is indicated by a dotted horizontal line. Data points positioned above this line are considered significant. Data points located on either side of the vertical line have log2 fold change values exceeding the set threshold of 2.0 or greater.

Leading edge gene analysis: The GSEA package (v4.3.2) provides a list of ranked genes, termed leading edge genes (LE), that have the most influence on enrichment significance for identified pathways [20]. This list is a subset of identified genes that contribute the most to the enrichment score for specific pathways within the gene set. We performed frequency analysis to identify genes shared across multiple enriched pathways, prioritizing genes that drive transcriptomic changes. This was expressed in tabular format.

Pathway analyses: Gene Set Enrichment Analysis (GSEA) was utilized to perform integrated pathway analysis of the 2 dpi and 5 dpi CVB3-iBEC infection transcriptomes [20]. GSEA determines the top up- and downregulated pathways of the dataset by identifying subsets of genes that are overrepresented in the transcriptomic data. Using a ranked gene list, sorted by *p*-value and log2 fold change, GSEA data were expressed in tabular format as well as ridge-plot and heatmap visualization. Next, EnrichR (v3.4) was used to examine the top 10% up- and downregulated genes and used a scoring system to identify pathways that are up- and downregulated, based on the differential gene expression list [45]. EnrichR results were expressed in tabular format. Unlike GSEA, which evaluates enrichment scores across the entire ranked gene list to detect subtle changes in expression, Enrichr performs an over-representation analysis limited to predefined subsets of DEGs. EnrichR is useful for identifying strongly enriched pathways that involve only the most dysregulated genes, while GSEA is better for whole transcriptome pathway analysis [20,45,49].

iLINCs: The LINCS (Library of Integrated Network-based Signatures) (v2.6.7) package was utilized to identify drugs and small-molecule perturbagens that exhibit signatures that are concordant (i.e., “simulate”) or discordant (i.e., “reverse”) with the iBEC transcriptome following CVB3 infection at both 2 and 5 dpi. By cross-referencing the L1000 mRNA drug signatures from the iLINCS database with the L1000 mRNA expression patterns extracted from the 2 and 5 dpi differential gene expression lists, we generated a list of drug perturbagens. iLINCs’ output was expressed in tabular format, with an iLINCs Pearson correlation threshold of concordance (>0.321) and discordance (<−0.321) to filter for drugs of interest [18]. The iLINCs threshold of >0.321 was adapted from previous work performed using with the iLINCs software (v2.6.7) [50,51].

Pheatmap (v1.0.13) and ggridges (v0.5.6): To show GSEA pathways and relative expression, the package “pheatmap” was used to produce heatmaps relative to NES and log2FC values derived from GSEA output [52]. The package filtered the top 75 GSEA pathways and identified that 39 of these pathways were shared across the 2 and 5 dpi (*p*adj < 0.05). A heatmap was created based on the normalized enrichment score (NES) between the shared pathways. The heatmap was generated with unsupervised hierarchical clustering of NESs. Ridge plots were created using “ggridges” to visualize relative mRNA expression among the top 39 GSEA pathways [53]. Log2 fold change was extracted for each gene in the core pathway enrichment list and plotted into a single data frame. The plots were then combined and color-coded according to dpi.

RDKit similarity matrix and UMAP plotting: To show structural similarity among drugs/small molecules from iLINCs output, SMILES-formatted molecular strings of the top 2 dpi discordant iLINCs perturbagens were converted to RDKit molecule objects and processed to generate Morgan fingerprints [54]. Pairwise Tanimoto similarity coefficients were calculated using RDKit’s DataStructs module [54]. A Tanimoto coefficient that approaches “1.0” indicates a higher chemical structure similarity between two molecules. K-means clustered Heatmaps were generated using the Seaborn package (v0.13.2) to visualize similarity matrices [55]. To complement this analysis, clustered Uniform Manifold Approximation and Projection (UMAP) plots were generated for fingerprint vectors and clustered using K-means clustering to visualize chemical space relationships among compounds [54]. Perturbagens of interest were labeled accordingly.

SwissTargetPrediction (http://www.swisstargetprediction.ch/citing.php; accessed on 24 June 2025) was used to target genes from *Homo sapiens* to examine possible molecular targets for our drugs of interest [35]. We extracted a list of all predicted binding targets and compared the unique and shared binding targets across all drugs of interest. We then developed a list of the top shared binding targets along with each drugs predicted binding probability using SwissTargetPrediction [35]. This was expressed as raw numeric values in a Venn diagram and shared gene targets were shown in tabular format.

## 5. Conclusions

We applied a modern, cutting-edge bioinformatics approach to re-analyze a previously published RNAseq dataset of CVB3 infection on iBECs using an integrated bioinformatics framework that combines GSEA, EnrichR, and iLINCs-based perturbagen analyses. We identified novel drug candidates and performed enriched pathway analysis to provide a more defined view of this model. Notably, we identified several candidate small-molecule inhibitors, including MEK and VEGF pathway modulators, with potential for repurposing as antiviral therapies. Further studies investigating CVB3 when interacting with an iBEC model are needed to show the efficacy of the drugs and small molecules found.

## Figures and Tables

**Figure 1 ijms-26-07041-f001:**
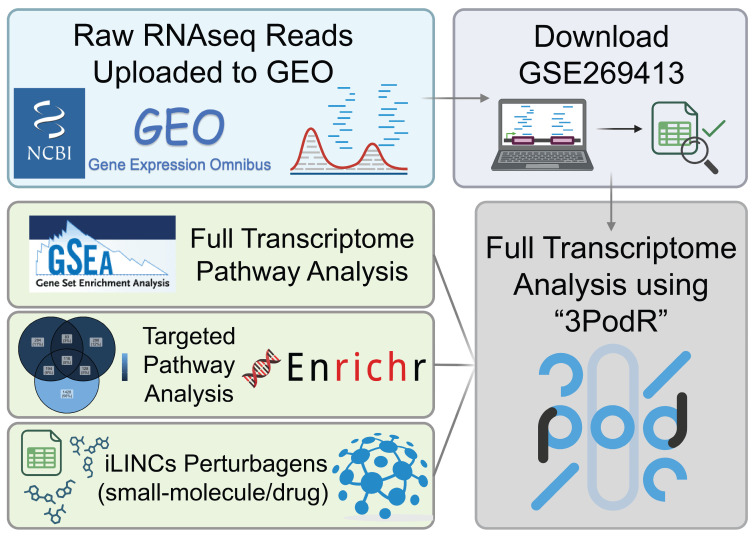
A previously published CVB-infected iBEC RNA sequencing dataset was queried and downloaded from Gene Expression Omnibus (GEO) (GSE269413). Whole transcriptome analysis was then run using a suite of bioinformatics tools in the “3PodR” package, including differential gene expression, pathway analysis, and perturbagen analysis [19]. Visualizations were subsequently generated from this analysis. The “3PodR” tool package (pre-release, https://doi.org/10.5281/zenodo.8190833, accessed on 8 August 2024) enabled the interpretation of transcriptomic changes, biomolecular pathways, and possible perturbagens were identified, providing a unique transcriptional analysis of this dataset. Created in Biorender. Jacob Wood. (2025) https://BioRender.com (accessed on 10 January 2025).

**Figure 2 ijms-26-07041-f002:**
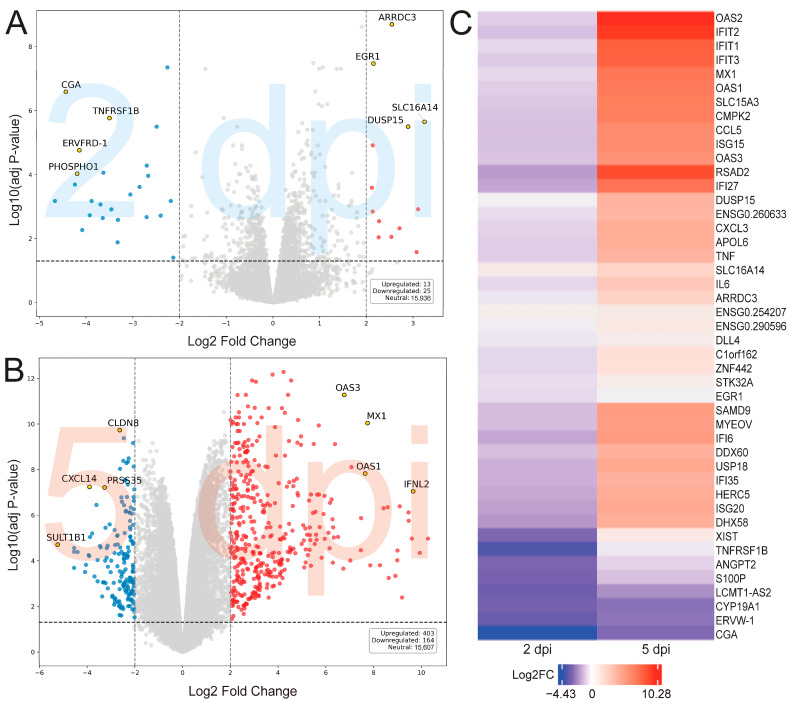
(**A**,**B**) Volcano plots representing differential gene expression at 2 (**A**) and 5 dpi (**B**), with genes color-coded and annotated. A threshold-adjusted *p*-value of *p*adj < 0.05 was used. The y-axis represents a −log10 *p*-value scale. The x-axis represents log2 fold change, the y-axis represents adjusted *p*-value. Upregulated genes are red dots; downregulated genes are blue dots. Highlighted genes are represented as yellow dots. (**C**) Heatmap showing the top differentially expressed genes in iBECs following coxsackievirus B3 (CVB) at 2 and 5 dpi. The left panel shows the log2 fold change in gene expression in cells infected at 2 dpi compared to mock-infected controls; the right panel shows gene expression changes at 5 dpi relative to their respective mock controls. Color intensity reflects log2 fold change values, with positive values indicating upregulation and negative values reflecting downregulation. Gene names are listed accordingly.

**Figure 3 ijms-26-07041-f003:**
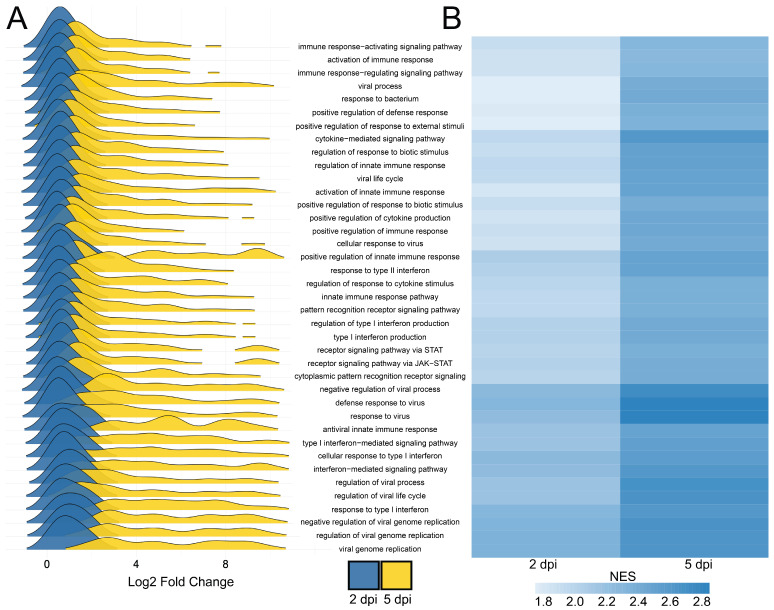
Gene Set Enrichment Analysis (GSEA) of differentially expressed genes for 2 and 5 dpi. (**A**) Ridge plots displaying the distribution of log2FC for DEGs in the top 39 significantly (*p*adj < 0.05) shared GSEA pathways between 2 and 5 dpi. These 37 pathways were selected by identifying an overlap among the top 75 GSEA pathways for 2 and 5 dpi. The x-axis displays log2 fold change values, while the y-axis lists GSEA pathway names corresponding to those in panels A and B. The ridges within the plot are colored, where blue is 2 dpi, and yellow is 5 dpi. (**B**) Heatmap showing the normalized enrichment scores (NES) for the top 39 significantly (*p*adj < 0.05) shared GSEA pathways. Pathways are clustered hierarchically by normalized enrichment scores (NES), with a color scale ranging from light blue (lower enrichment) to dark blue (higher enrichment).

**Figure 4 ijms-26-07041-f004:**
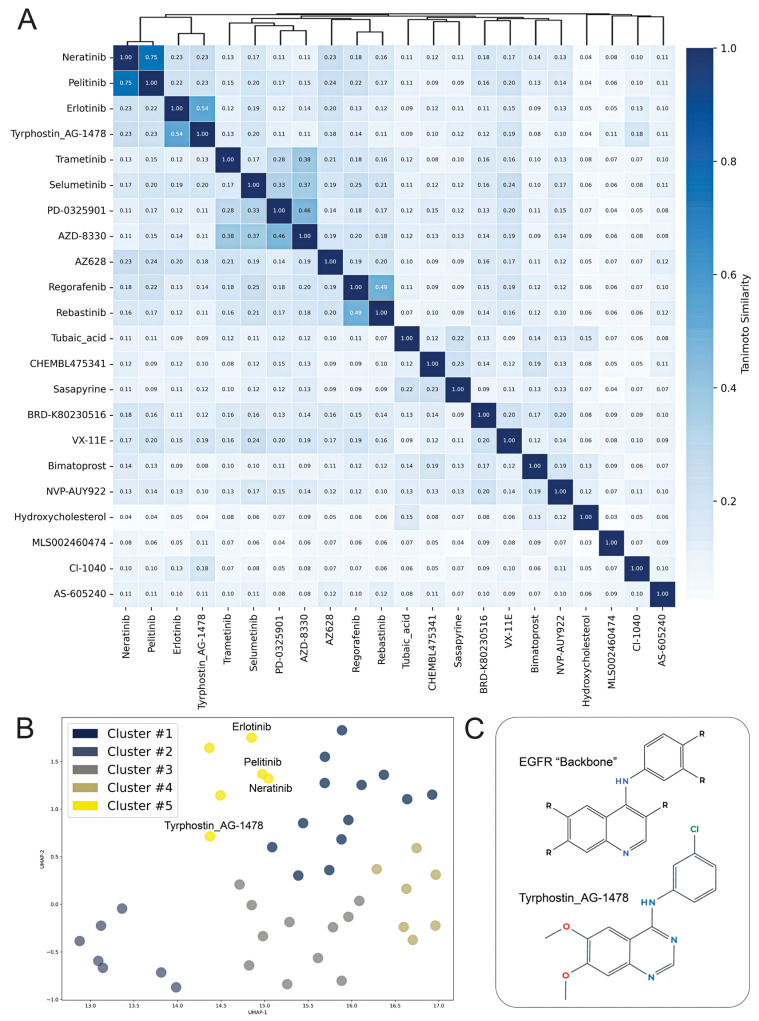
(**A**) Hierarchically clustered Tanimoto similarity heatmap comparing chemical structures of the top 22 candidate compounds identified at 2 dpi by iLINCs. Each cell represents the paired Tanimoto coefficient between paired compounds, with higher values indicating greater structural similarity, with a color scale representation. (**B**) UMAP plot visualizing the compounds from 2 and 5 dpi based on 5-group Kmeans clustering of Tanimoto coefficients, highlighting structurally related molecules, with 4 drugs labeled as followed: erlotinib, pelitinib, neratinib, and tyrphostin_AG-1478. (**C**) Representative chemical scaffold (“Backbone”) shared among EGFR inhibitors (erlotinib, pelitinib, and neratinib), as well as Tyrphostin_AG-1478’s chemical structure, indicated as structurally similar top hits at 2 dpi.

**Figure 5 ijms-26-07041-f005:**
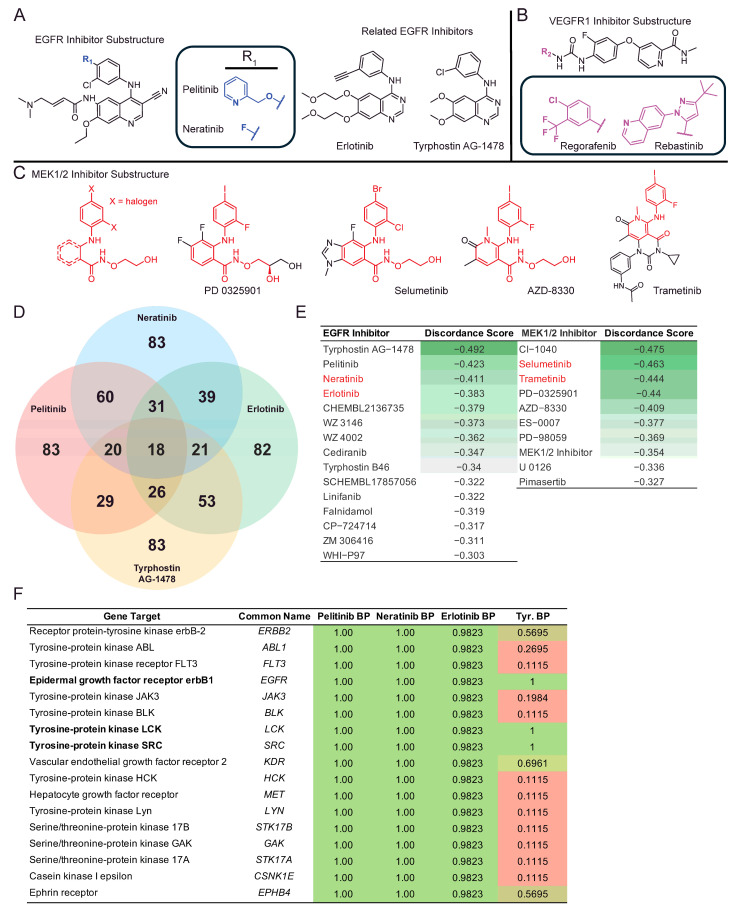
(**A**) Substructure of closely related EGFR inhibitors and related analogs. (**B**) Common structures of an additional cluster of RTK inhibitors. (**C**) Common substructure of MEK inhibitors is shown in red. Trametinib is also shown, which clusters with the MEK1/2 inhibitors based on Tanimoto coefficient. (**D**) Four-way Venn diagram showing neratinib, pelitinib, tyrphostin AG-1478, erlotinib and their numerical value for unique and shared SwissTargetPrediction docking gene targets. (**E**) Tables from iLINCs discordant drug signatures at 2 dpi showing EGFR and MEK1/2 inhibitors, with FDA-approved drugs highlighted in red text, and corresponding discordant score color scale in green. Abbreviations: Tyr: tyrphostin AG-1478. (**F**) Table of 18 SwissTargetPrediction docking gene targets shared across neratinib, pelitinib, tyrphostin AG-1478, and erlotinib, with docking probability scores with representative green and red color scale.

**Table 1 ijms-26-07041-t001:** Previous studies.

Species	Cell Line	Significant Genes	Significant Pathways	Methods	Authorship
CVB3	HeLa	CCNG2, GADD45B, PIM1, RBM15, KLF10	FoxO	DESeq2, PPI network, KEGG	Liu et al. [9]
CVB3	HeLa/HepG2	IL3RA, CASP3, ATF1, TPM1, MMP9, CCL27, TNFa	IFN and IL	DNA Microarray	Rassmann et al. [10]
CVB3	A/J (H-2)	PABP, UBP41, inorganic pyrophosphatase genes	B-Ox. And CAC/ETC pathways	DNA Microarray	Taylor et al. [11]
CVB3	nHPC’s	RIG-I, MDA5, IFN-, IP-10, ISG15, OAS1, OAS2	Innate Immune response/IFN	Quantseq, KEGG	Oh et al. [12]
CVB3	CMEC	ERK1/2, FKN, CX3C	MAPK, ERK1/2	Western blotting, RNAi	Wen et al. [13]

Legend: Table showing previous literature investigating CVB3 and RNAseq, classified by cell line infected and methods used for sequencing. Abbreviations: B-Ox.; Beta-Oxidation, CAC/ETC; Citric Acid Cycle/Electron Transport Chain.

**Table 2 ijms-26-07041-t002:** GSEA upregulated pathways 2 and 5 days post infection.

2 Day GSEA Pathways	5 Day GSEA Pathways	*p*adj	Rank
Regulation of viral genome replication	**Response to virus**	<0.01	1
Negative regulation of viral genome replication	**Defense response to symbiont**	<0.01	2
**Negative regulation of viral process**	**Defense response to virus**	<0.01	3
**Defense response to symbiont**	Defense response to other organism	<0.01	4
**Defense response to virus**	Innate immune response	<0.01	5
Regulation of viral life cycle	Response to external biotic stimulus	<0.01	6
Positive regulation of IFN-beta production	Response to other organism	<0.01	7
**Response to virus**	Response to biotic stimulus	<0.01	8
Positive regulation of mirna process	**Negative regulation of viral process**	<0.01	9
Regulation of viral process	Immune response	<0.01	10

**Legend:** Table of the top 10 2 and 5 dpi GSEA pathways, sorted by ranking (1–10) based on NES, and *p*adj value; shared 2 and 5 dpi pathways are shown in bold. Abbreviations: IFN, Interferon.

**Table 3 ijms-26-07041-t003:** Discordant perturbagens 2 days post infection.

Perturbagen	Discordance Score	Mechanism of Action
N6022	−0.5	Unknown
Tyrphostin AG-1478	−0.49	EGFR inhibitor
CI-1040	−0.47	MEK inhibitor
BRD-K80230516	−0.47	Unknown
Selumetinib	−0.46	MEK inhibitor
Tubaic acid	−0.45	Mitochondrial complex I inhibitor
AZ628	−0.44	RAF inhibitor
Trametinib	−0.44	MEK inhibitor
PD-0325901	−0.44	MAP kinase, MEK inhibitor
Bimatoprost	−0.42	Unknown

Legend: Table of top 10 discordant perturbagens at 2 dpi, sorted by discordance score.

**Table 4 ijms-26-07041-t004:** Discordant perturbagens 5 days post infection.

Perturbagen	Discordance Score	Mechanism of Action
Hydroquinidine	−0.46	Antiarrhythmic
AZD-8055	−0.45	MTOR inhibitor
BMS-536924	−0.42	IGF-1 inhibitor
TSU-68	−0.42	FGFR, PDGFR, VEGFR inhibitor
CAY-10594	−0.41	Phospholipase inhibitor
Pilocarpine	−0.40	Acetylcholine receptor agonist
Ampalex	−0.40	Unknown
BRD-K42573370-001-01-1	−0.40	Nucleophosmin inhibitor
Betahistine	−0.40	Histamine receptor agonist/antagonist
Naftifine	−0.4	Squalene monooxygenase inhibitor

Legend: Table of top 10 discordant perturbagens at 5 dpi, sorted by discordance score.

## Data Availability

The original data presented in the study are openly available in the Gene Expression Omnibus at GSE269413 (https://www.ncbi.nlm.nih.gov/geo/query/acc.cgi?acc=GSE269413; accessed on 8 August 2024).

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
