# Peer review of "Identification of Drug Repurposing Candidates for Coxsackievirus B3 Infection in iPSC-Derived Brain-like Endothelial Cells"

_ijms, 2025, doi:10.3390/ijms26157041_

Round 1

Reviewer 1 Report

Comments and Suggestions for Authors

The manuscript: "Identification of Drug Repurposing Candidates for Cox-sackievirus B3 Infection in iPSC-Derived Brain-Like Endothelial Cells", presents a modern bioinformatics approximation to analyze a RNAseq dataset of CVB3 infection based on iBECs model.

The manuscript is well-written, and the methodology employed is both modern and clearly described. However, given the manuscript's title, a greater emphasis on the identification of drug-repurposing candidates would have been expected. If the researchers' primary aim was to utilize bioinformatics tools exclusively, additional bioinformatics approaches could have been applied to confirm or elucidate the potential mechanisms of action of the identified candidates. Therefore, it is recommended that the authors expand the study in this regard with docking and molecular dynamics studies.

Author Response

In reference to the article titled: Identification of Drug Repurposing Candidates for Coxsackievirus B3 Infection in iPSC-Derived Brain-Like Endothelial Cells

Introduction: Thank you very much for taking the time to review this manuscript. Please find the detailed responses below and the corresponding revisions/corrections in the re-submitted files. We are very excited for this manuscript to be published with MDPI-IJMA and look forward to hearing from you.

Reviewer 1:

Comment #1: “However, given the manuscript's title, a greater emphasis on the identification of drug-repurposing candidates would have been expected. If the researchers' primary aim was to utilize bioinformatics tools exclusively, additional bioinformatics approaches could have been applied to confirm or elucidate the potential mechanisms of action of the identified candidates. Therefore, it is recommended that the authors expand the study in this regard with docking and molecular dynamics studies.”

Response #1: Thank you for pointing this out. We agree with this comment and thought that more analysis could go into docking studies for the drugs that we identified as potential repurposing candidates. Therefore, we have provided an updated Figure 5, which shows the 4 drugs of interest that we selected as top discordant signatures and the unique/overlapping docking simulation results as a Venn diagram (Figure 5C), and the expanded docking targets as a table (Figure 5D). One of these drugs (tyrphostin AG-1478) was used previously in a enterovirus-71 study, highlighting outside work that has been done to show efficacy of these EGFR inhibitors. We believe this highlights potential genetic targets that these drugs may act on, in addition to their pharmaceutical mechanism of action. (lines 272-280)

          In addition to this, we have also pulled the top EGFR and MEK1/2 inhibitors out from the dataset, highlighting their FDA approval (in red) and their discordance score with the dataset (green color scale). This highlights why these drugs were chosen for our docking study with EGFR inhibitors, showing that FDA approved drugs appeared in our dataset as discordant signature’s that are viable for docking studies along with other experimental drugs.

Reviewer 2 Report

Comments and Suggestions for Authors

The manuscript presents an original concept for the treatment of enterovirus infections based on an analysis of the effect of Coxsackie B3 virus on the blood-brain barrier. Based on published data on RNA sequences, human brain stem-endothelial cells infected with the virus were studied. Through transcriptomic analysis, an original approach for discovering new applications of drugs as antivirals is proposed. Pelitinib, neratinib and related structures are indicated as candidates. The authors use a complex of precise methods, including differential gene expression and leading edge gene analyses, pathway analyses, LINCS package, pheatmap, RDKit matrix and UMAP plotting. The results obtained and the analysis on them are an original contribution to the therapy of enterovirus infections, complementing the use of inhibitors of viral replication.

Author Response

In reference to the article titled: Identification of Drug Repurposing Candidates for Coxsackievirus B3 Infection in iPSC-Derived Brain-Like Endothelial Cells

Introduction: Thank you very much for taking the time to review this manuscript. Please find the detailed responses below and the corresponding revisions/corrections in the re-submitted files. We are very excited for this manuscript to be published with MDPI-IJMA and look forward to hearing from you.

Reviewer 1:

Comment #1: “However, given the manuscript's title, a greater emphasis on the identification of drug-repurposing candidates would have been expected. If the researchers' primary aim was to utilize bioinformatics tools exclusively, additional bioinformatics approaches could have been applied to confirm or elucidate the potential mechanisms of action of the identified candidates. Therefore, it is recommended that the authors expand the study in this regard with docking and molecular dynamics studies.”

Response #1: Thank you for pointing this out. We agree with this comment and thought that more analysis could go into docking studies for the drugs that we identified as potential repurposing candidates. Therefore, we have provided an updated Figure 5, which shows the 4 drugs of interest that we selected as top discordant signatures and the unique/overlapping docking simulation results as a Venn diagram (Figure 5C), and the expanded docking targets as a table (Figure 5D). One of these drugs (tyrphostin AG-1478) was used previously in a enterovirus-71 study, highlighting outside work that has been done to show efficacy of these EGFR inhibitors. We believe this highlights potential genetic targets that these drugs may act on, in addition to their pharmaceutical mechanism of action. (lines 272-280)

          In addition to this, we have also pulled the top EGFR and MEK1/2 inhibitors out from the dataset, highlighting their FDA approval (in red) and their discordance score with the dataset (green color scale). This highlights why these drugs were chosen for our docking study with EGFR inhibitors, showing that FDA approved drugs appeared in our dataset as discordant signature’s that are viable for docking studies along with other experimental drugs.

Reviewer 2:

Comment #1: “The manuscript presents an original concept for the treatment of enterovirus infections based on an analysis of the effect of Coxsackie B3 virus on the blood-brain barrier. Based on published data on RNA sequences, human brain stem-endothelial cells infected with the virus were studied. Through transcriptomic analysis, an original approach for discovering new applications of drugs as antivirals is proposed. Pelitinib, neratinib and related structures are indicated as candidates. The authors use a complex of precise methods, including differential gene expression and leading-edge gene analyses, pathway analyses, LINCS package, pheatmap, RDKit matrix and UMAP plotting. The results obtained and the analysis on them are an original contribution to the therapy of enterovirus infections, complementing the use of inhibitors of viral replication.”

Changes Made #1: Thank you very much for taking the time to read over this manuscript and provide feedback. No changes were made per request of reviewer.

Additional comments (email):

Editor Comment #1: “We saw "Jacob F. Wood" as the corresponding author in the document you submitted, but on the submission system, "Robert E. McCullumsmith" was marked as the corresponding author. Please confirm the correct list of corresponding authors with us as soon as possible.”

Response #1: Thank you for this comment. We have changed the corresponding author in the manuscript file to match the submission corresponding author. This can be found on Page 1 of the main manuscript under “Correspondence”.

Editor Comment #2: Duplicate references: During our inspection, we found that the following references seem to be repeated. Please carefully check and make corrections. ref 21=>27

Response #2: Thank you for pointing this reference error out. We have removed the citation reference to cite #27, as this is the same as the citation reference to cite #21. I have also renumbered the citations to match the order to the methods being last, which was previously incorrect. The use of the citations is now consistent throughout the main manuscript.

Additional Clarifications:

Clarification #1: We have changed the color scale to a more neutral blues color for figure 3B heatmap, given the possible misleading color scale (red to blue) which could be indicating down and upregulation much like the heatmap in figure 2A. This figure scale starts at 1.8 which would not be considered “downregulation”

Clarification #2: We have also added a discussion sentence (lines 256-257) highlighting why picking 2 dpi would be crucial for drug identification in reference to figure 4 A-C. This further clarifies the methodology for selection of our drug repurposing candidates.

Once again, thank you for taking the time to review this manuscript. All comments were extremely valuable and enhanced the academic rigor of this publication. We look forward to hearing from you.

Jacob F. Wood

6/19/2025